# Electromagnetic-Thermal Coupling Study for RF Compression Cavity Applied to Ultrafast Electron Diffraction

**DOI:** 10.3390/s23177455

**Published:** 2023-08-27

**Authors:** Zhen Wang, Jian Xu, Xintian Cai, Zhiyin Gan, Caoyue Ji, Cheng Lei, Sheng Liu

**Affiliations:** 1The Institute of Technological Sciences, Wuhan University, Wuhan 430072, China; wang.zhen@whu.edu.cn (Z.W.); caixintian@whu.edu.cn (X.C.); 2018106520021@whu.edu.cn (C.J.); 2School of Electrical & Electronic Engineering, Wuhan Polytechnic University, Wuhan 430023, China; jianxjlj@whpu.edu.cn; 3School of Mechanical Science & Engineering, Huazhong University of Science & Technology, Wuhan 430074, China; ganzhiyin@hust.edu.cn; 4School of Power & Mechanical Engineering, Wuhan University, Wuhan 430072, China

**Keywords:** dynamic atomic motion, RF compression cavity, ultrafast electron diffraction

## Abstract

Ultrafast electron diffraction (UED) is a powerful tool for observing the evolution of transient structures at the atomic level. However, temporal resolution is a huge challenge for UEDs, mainly depending on the pulse duration. Unfortunately, the Coulomb force between electrons causes the pulse duration to increase continually when propagating, reducing the temporal resolution. In this paper, we theoretically design a radio frequency (RF) compression cavity using the finite-element method of electromagnetic–thermal coupling to overcome this limitation and obtain a high-brightness, short-pulse-duration, and stable electron beam. In addition, the cavity’s size parameters are optimized, and a water-cooling system is designed to ensure stable operation. To the best of our knowledge, this is the first time that the electromagnetic–thermal coupling method has been used to study the RF cavity applied to UED. The results show that the RF cavity operates in TM010 mode with a resonant frequency of 2970 MHz and generates a resonant electric field. This mode of operation generates an electric field that varies periodically and transiently, compressing the electronic pulse duration. The electromagnetic–thermal coupling method proposed in this study effectively improves the temporal resolution of UED.

## 1. Introduction

After decades of research and development, ultrafast electron diffraction (UED) has provided us with the technical tools to study ultrafast phenomena [1,2,3]. Compared to conventional electron diffraction tools, UED incorporates temporal resolution, which combines the ultra-high spatial resolution of electron diffraction with the temporal requirements for studying ultrafast dynamics. The application of UED has contributed to many discoveries of ultrafast phenomena in physics [4,5,6], chemistry [7,8,9,10], and biology [11]. It has become a powerful tool for revealing the structural dynamics of molecules and materials at the atomic level.

In ultrafast electron diffraction (UED), temporal and spatial resolution are the two most important metrics. A higher brightness and shorter-pulse-duration probes are required to study kinetic processes with shorter onset times. Scientists have proposed several schemes to improve the temporal and spatial resolution. For example, some research groups have used compact electron cavities [12,13,14,15,16,17,18] to reduce the size of the electron cavity as much as possible, thus reducing the transmission time. However, the size that can be reduced is limited due to the presence of magnetic and deflection coils, which limits improvement in time resolution. Another method is to increase the acceleration voltage to improve the temporal resolution [19,20,21,22,23], which is due to the fact that the higher the acceleration voltage, the higher the electron beam acceleration and the smaller the spread of the beam during propagation. The acceleration voltage of MeV has been achieved, but the construction cost is high. Electron pulse compression [24,25,26,27,28,29] is another method that compresses the electron beam during its propagation, allowing for joint experimental probing with material growth equipment, reducing the damage caused by oxidative contamination of the material in the air and allowing for better visualization of the material properties themselves. B.J. Siwick et al., through calculations, found that the Coulomb interaction between electrons causes the electron pulse to widen, forming a linear velocity chirp [30]. Although researchers have made remarkable progress in UED over the past few decades, there is still room for improvement in temporal resolution. However, the electromagnetic–thermal coupling of the RF cavity is still lacking in the field of study.

Based on Siwick’s theory, applying an RF compression cavity to compress the longitudinal pulse width of ultrafast electrons is studied [12]. A 60 keV DC voltage accelerates the electrons, and the position of the RF compression cavity is determined by the distance of the sample stage from the cathode of the electron gun. A time-varying electric field is applied to flip the chirp of the electron beam as it passes through the RF cavity, ensuring that the energy distribution of the electron beam is reversed and the electron beam pulse width is compressed longitudinally. This paper focuses on the design optimization of the RF compression cavity. An RF compression cavity system is designed based on the femtosecond (fs) laser parameters and the distance from the electron gun to the sample stage. Our ultrafast electron diffraction system enables the compression of the electron pulse width, laying the foundation for improving the temporal resolution and providing a possible research direction.

## 2. Ultrafast Electron Diffraction System

Ultrafast electron diffraction (UED) is a pump–probe technology that uses a pulsed laser to irradiate the surface of a sample and an electron beam as a probe to detect dynamic processes at the atomic level. UED plays a vital role in surface imaging. The electron beam is incident on the sample surface at a grazing angle of 1–5°. We aim to increase the brightness of the electron beam cluster. The increase in brightness of the electron beam cluster implies an extension of the electron beam pulse width and a decrease in time resolution. To reduce the time resolution, we use the RF compression cavity to compress the longitudinal width of the electron beam cluster. Our fs laser has a pulse width of 30 fs and a repetition frequency of 110 MHz. According to our designed RF compression cavity electric field and control system of electron beam synchronization, the repetition frequency of the laser is 110 MHz, and then the resonant frequency of the RF compression cavity is 2970 MHz.

The overall schematic diagram of the electron gun is shown in Figure 1a. The pulsed laser, with a wavelength of 266 nm and a pulse width of 30 fs, enters from the left flange window and passes through the 40 nanometer-thick silver thin film photocathode. Due to the photoelectric effect, the pulsed laser generates femtosecond-scale electron pulses when it hits the film. The generated electron beam enters the DC acceleration zone at 60 keV with an acceleration distance of 9 mm. Finally, the electron beam with an energy of about 0.1 pC (i.e., 10^6^ electrons) is accelerated to about one-third of the speed of light. The electron beam then passes through the central hole of the anode plate with a diameter of 0.1 mm, the magnetic lens, the deflection plate, the RF compression cavity, and the sample cavity. Finally, the diffraction information is obtained in the imaging system. The magnetic lens is used to focus the electron beam transversely to increase the brightness of the spot, while the deflection plate is used to adjust the position of the electron beam in the x- and y-directions, allowing the electron beam to find the sample position accurately. The RF compression cavity performs longitudinal compression of the electron beam to improve the time resolution.

Due to the space-charge effect, the longitudinal electron pulse width of the electron beam rapidly broadens as it propagates, forming a linear velocity chirp in the direction of electron beam propagation [30]. Figure 1b shows the simulation analysis of the longitudinal pulse width variation in the electron beam as it propagates, using the particle tracer software GPT (general particle trace) [31]. The solid black line represents the continuous broadening of the electron beam pulse width due to the space-charge effect when the RF cavity is not added. The red dashed line shows the simulation plot of the electron beam’s pulse width change during propagation after adding the RF cavity. By changing the magnitude of the RF compression cavity electric field, we can compress the longitudinal pulse width of the electron beam, allowing it to propagate to the sample stage with the shortest pulse width. According to the simulation results, when the electric field at the neck of the RF cavity is 1.45 MV/m, the electron beam flies to the sample stage with the shortest pulse width and the best pulse width compression effect. The compression distance is 150 mm at this time, which meets the overall design requirements. We set perfect boundary conditions in our simulation using GPT, and via calculation, we increased the electric design field strength of the RF compression cavity to 6.5 MV/m, as reported in the literature [32]. Based on the above calculations and analysis, we obtain the resonant frequency of the RF compression cavity as 2970 MHz and the electric field of the neck as 6.5 MV/m.

## 3. RF Compression Cavity System

Our designed RF compression cavity operates in the TM010 mode, generating a resonant electric field at the cavity neck position to provide a time-varying energy chirp to the longitudinally propagating electron beam, thereby compressing the electron beam pulse width. The RF compression cavity can be considered as a lens during the propagation of the electron beam, with the focal length given using the following equation [23]:(1)$$Z_{foucus}\approx \frac{2\sqrt{2/mU_{k}^{3/2}}}{ewd_{cav}E_{0}\cos\left(\theta \right)}\left[1+\frac{ed_{cav}E_{0}}{U_{k}}\sin\left(\theta \right)\right]$$
(2)$$E\left(t\right)=E_{0}\sin\left(\omega t-\frac{\omega d_{cav}}{2v_{c}}+\theta \right)$$
where *m* is the electron mass, *U_k_* is the kinetic energy of the electron pulse, *e* is the electron electric charge, *θ* is the angular frequency of the RF field, *d_cav_* is the effective cavity length of the RF cavity, and *ω* is the phase of the RF field.

A spatial electric field simulation diagram of the RF compression cavity is obtained using CST STUDIO SUITE simulation. Figure 2a–c in the graph represent the interaction between the electron beam and the electric field collection area in the RF cavity at different times. The red ellipse’s size denotes the electron beam’s pulse width. The green arrow in the cavity indicates the direction of the electric field. Figure 2a–c correspond to the three states of operation of the RF cavity. As shown in Figure 2a, when the faster electrons at the front of the electron beam enter the RF cavity, the electric field in the cavity is in the same direction as the electron propagation, which exerts a reverse force on the electron beam to slow it down. Since the electric field strength is time-varying, the strength of the electric field decreases linearly as the electrons propagate. As shown in Figure 2b, the electric field strength in the focusing region of the cavity decreases to zero as the center of the electron pulse propagates toward the center of the cavity. As shown in Figure 2c, when the second half of the electron pulse enters, the direction of the electric field in the cavity is opposite to the direction of electron propagation, which exerts an accelerating force on the electron beam. Therefore, the faster electrons in the first half of the electron beam are slowed down by applying an electric field in the same direction as the electrons are propagating, as shown in Figure 2a. Conversely, electrons with a slower velocity at the rear of the electron beam accelerate when an electric field is applied in the opposite direction to that in which the electrons are propagating (Figure 2c). Figure 2d shows SOLIDWORKS’s 3D design schematic of the RF compression cavity. Figure 2e shows the physical diagram of the RF compression cavity, and the asymmetric holes in the figure are used for positioning and alignment during welding, but there are no specific requirements for their location.

## 4. Electromagnetic–Thermal Coupling Study

### 4.1. TM Mode in the Cylindrical Resonant Cavity

We start with a standard resonant cavity design. According to the principle of electron beam compression, we need the electric field in the RF cavity to be mainly concentrated in the compressed region and the magnetic field to be far away from the compact area, so the TM010 mode is chosen. In the cylindrical coordinate system, the analytic solution of the cylindrical RF compression cavity can be expressed as [33]:(3)$$\frac{\partial^{2}E_{z}}{\partial z^{2}}+\frac{1}{r}\frac{\partial E_{z}}{\partial r}+\frac{\partial^{2}E_{z}}{\partial r^{2}}-\frac{1}{c^{2}}\frac{\partial^{2}E_{z}}{\partial t^{2}}=0$$
where *E_z_* is the component of the electric field in the z-direction, *z* is the direction of electron propagation, and *t* is the time.

At the circumference in the cylindrical resonant cavity, the boundary condition is that the tangential component of the electric field is zero, and the separation variable is solved to obtain *E_z_*(*r*,*z*,*t*). Then, in the working mode of the TM010, the longitudinal electric field E and the angular magnetic field B, according to Maxwell’s equations, can be expressed as [32]:(4)$$\left\{\begin{array}{c} E_{z}\left(r,t\right)=E_{0}J_{0}\left(\frac{x_{01}}{R}r\right)\cos\left(\omega t+\varphi_{c}\right)\\ B_{\theta }\left(r,t\right)=-\frac{E_{0}}{c}J_{1}\left(\frac{x_{01}}{R}r\right)\sin\left(\omega t+\varphi_{c}\right) \end{array}\right.$$
where *x*_01_ ≈ 2.405, *c* is the speed of light, and *φ_c_* is the phase of the electric field in the RF cavity. To compress the effect, we should ensure that the electric field is at the maximum in the axial position and the magnetic field is zero; then, *J*_0_(*r* = *R*) = 0 and *J*_0_(*r* = 0) = 1. Since the electron pulse is excited by the laser, the electron pulse and the laser pulse have the same frequency and phase, and the electric field in the cavity should be kept synchronized with the phase of the laser pulse. According to the compression principle, when the center of the electron passes through the center of the compressed electric field, the electric field intensity is zero. According to this principle, we can adjust the magnitude of the phase to achieve the desired direction of the electric field when the electron beam enters the RF compression cavity.

### 4.2. Design Optimization of Ω RF Cavity

Since the cylindrical RF cavity requires a very high input power to meet the design requirements, the design of the RF signal source is quite tricky. Therefore, we use the optimized Ω cavity. Due to the irregular shape of the Ω cavity, obtaining the cavity’s internal field intensity distribution, quality factor, and resonance frequency using theoretical calculation is impossible. Therefore, we designed the cavity for the first time using COMSOL Multiphysics to obtain the resonant frequency, electric field intensity, and quality factor to meet the design requirements. In this paper [32], the authors designed the RF compression cavity using the 2D software POISSON SUPERFISH [34] and fabricated the cavity for experiments. The experimental results are consistent with the simulation results, proving the reliability of the POISSON SUPERFISH simulation results. To confirm the reliability of our design, we compare the results of the COMSOL Multiphysics simulation with the results of the POISSON SUPERFISH simulation. Using COMSOL Multiphysics, we can simulate the resonant frequency, electric field strength, quality factor, and the RF compression cavity’s electric, magnetic, and thermal conductivity.

After designing the shape of the Ω cavity according to the previous study [32], the cavity is dimensionally optimized to achieve the required resonant frequency of 2970 MHz, and the optimization results are shown in Figure 3.

Figure 3a shows the cavity dimensions corresponding to an optimized resonant frequency of 2.97035 GHz, and Figure 3b shows the relationship between the resonant frequencies by changing the cavity dimensions during the optimization process. Our simulation is based on a cavity size corresponding to 2970 MHz with a size variation of ±80 µm. The results show that the corresponding frequency changes from 2976.8 MHz to 2963.2 MHz.

The resonant frequency is determined based on the repetition frequency of the fs laser and our photoelectric synchronization control system. At present, the repetition frequency of the fs laser is 110 MHz, and the resonant frequency is allowed to vary in the interval from 2968 MHz to 2974 MHz depending on the type of voltage-controlled oscillator in our synchronous control system. Based on the requirements of the frequency range and the simulation calculations results, our cavity’s manufacturing accuracy is 10 μm.

Figure 4 shows the results of the COMSOL Multiphysics simulation of the design frequency at 2.97035 GHz. From the simulation results, *E_z_*max(*r* = 0) = 7.3 MV/m, *Q* = 9196, and Power = 455 W. Figure 4b–d represent the longitudinal electric field strength, transverse electric field strength, and magnetic field strength normalized to *E_z_*(*r* = 0, *z* = 0) at different values of r-direction to the axial position z-direction where *r* is denoted as the radial position. The working range of the RF cavity we designed is between −2 mm and 2 mm in the z-direction and −2 mm and 2 mm in the r-direction. As the cavity is an axisymmetric structure, the internal field distribution of the cavity is also axisymmetric in the r direction. Therefore, we only need to analyze the range of 0 to 2 mm in the r direction. We divided it into four regions for analysis.

Figure 4a shows the evolution of the electric field in the interval from −10 mm to 10 mm in the z-direction for r = 0. Figure 4b shows the ratio of Ez, the electric field in the z-direction at different r positions to Ezmax, and the maximum value of Ez at r = 0. It can be seen from the figure that the change in the electric field at different r positions in the working area we are concerned about, which is the range of −2 mm to 2 mm in the z-direction, is small, indicating a uniform distribution of electric field in this area, which can exert a constant electric field force on the electron beam. Figure 4c shows the variation trend of Er, the electric field component in the r-direction, which mainly exerts a transverse force on the electron beam in the r-direction. It can be seen from the figure that, in the range of −2 mm to 2 mm in the z-direction that we are concerned about, the magnitude of Er is almost zero, indicating a negligible effect on the focusing or divergence of the electron beam. We can adjust the focusing lens to compensate for this effect and achieve the best result for the electron beam reaching the sample stage. In the RF cavity, electric and magnetic fields are generated, and the magnetic field will change the flight direction of electrons. Figure 4d shows the variation in the magnetic field force perpendicular to the direction of electron propagation in the working area. It can be seen from the figure that the magnetic field force is almost zero when r = 0, and the ratio of the maximum magnetic field force at r = 1.8 to the maximum electric field force is only 0.04. From these results, we can conclude that the effect of magnetic field force on the electron beam in the working area can be ignored. The RF cavity has multiple resonant frequencies, so we use COMSOL Multiphysics to explore their resonant frequencies. In the TM010 mode, the electric field of the RF compression cavity mainly concentrates at the neck of the cavity, and the resonant frequency is 2.97035 GHz. The adjacent resonant frequency is 4.75295 GHz, which differs significantly from the resonant frequency, reducing the frequency selection error caused by the proximity of the resonant frequencies.

Previously, two-dimensional software POISSON SUPERFISH has been predominantly utilized for simulating radio frequency (RF) cavities. Cavity designs generated through this software have been widely applied with a high level of proficiency. To enhance our research, we intend to employ COMSOL to simulate, verify, and calculate the newly designed RF cavity model and utilize POISSON SUPERFISH to validate simulation and calculations. Figure 5 shows the simulation results using the commonly used 2D software POISSON SUPEREFISH. Comparing the simulation results in Figure 5a–d with those in Figure 4a–d, it can be seen that the simulation results are similar, which proves that the COMSOL Multiphysics simulation results can be accurate and reliable. Figure 5e shows the electric field strength distribution in the RF cavity, and Figure 5f shows the magnetic field strength distribution in the RF cavity. The difference between the COMSOL Multiphysics simulation value (2.97035 GHz) and the simulation result (2.97065 GHz) calculated via the 2D simulation software POISSON SUPERFISH is only 300 KHz. The other results are shown in Table 1. Therefore, COMSOL Multiphysics can be used for electromagnetic and thermal multiphysics field simulation, providing results closer to the actual working conditions, thus reducing the design time and saving manufacturing costs.

Based on the comparative results, we can see that the calculation results of the two pieces of software are not significantly different. Furthermore, from this perspective, we can also prove that the cavity we design with COMSOL Multiphysics is feasible. The modeling in POSSION SUPERFISH is relatively cumbersome, and detailed modeling is prone to errors. Additionally, the power input is represented by a point, which is not a good simulation of the actual operation of the power input connector. The COMSOL Multiphysics design can effectively address these issues and simulate the electromagnetic, thermal, and multiphysics field at the same time.

Based on the simulation, our experiment has completed the cavity’s processing and manufacturing and the resonant frequency test; the test results are shown in the Figure 6. The experimental resonant frequency is 2.9738 GHz, consistent with the simulation with a 0.0038 GHz difference.

### 4.3. Thermal Simulation of RF Compression Cavity

When using an RF cavity, the temperature of the cavity will increase due to power dissipation. The increase in temperature will change the shape of the cavity. A change in the shape of the cavity causes a change in the resonant frequency. We can estimate the effect of temperature rise on the cavity frequency by the linear expansion coefficient *K_T_* [35].
(5)$$f_{0}=\frac{1}{\sqrt{\varepsilon_{0}\mu_{0}}R}x_{01}$$
(6)$$K_{T}=\frac{1}{R}\frac{dR}{dT}$$

The relationship between temperature and frequency is given by:(7)$$\frac{df}{dT}=\frac{-x_{01}k_{T}}{2\pi R\sqrt{\mu \varepsilon }}=f_{0}k_{T}$$

We plan to use oxygen-free copper processing for our RF compression cavity because copper has good electrical and thermal conductivity. The linear expansion coefficient of copper *K_T_* = 16.9 × 10^−6^ K^−1^, the frequency is 2.970 GHz, and the *df/dT* ≈ −50 kHz/K.

We design a water-cooling device to cope with the temperature rises in the cavity. As shown in Figure 2d,e, the water-cooled cavity is located in the area on both sides of the RF cavity, and we use COMSOL to perform electromagnetic and thermal multiphysics field simulation calculations for this process. We have chosen the top point of the *r* = 0.5 mm semicircle, as shown in Figure 3a, as the temperature measurement point. Figure 7a,b show the change in room temperature of the RF compression cavity after 2 h of operation at 2 kW injection power and an ambient temperature of 293 K before and after adding the water-cooling system. As can be seen from Figure 7a, the cavity temperature rises from 293 K to approximately 335 K without the water-cooling system. However, we can see from Figure 7b that, after adding the water-cooling system, the cavity temperature rises by about 0.6 K after two hours of operation, which shows that the effect of the water-cooling system in this process is pronounced.

Figure 7a,b show three-dimensional simulations of temperature variations. Figure 7c,d show the changes in the RF compression cavity’s working temperature before and after adding the water-cooling system. 

Figure 7c shows the simulation results of the working temperature field of the cavity in COMSOL Multiphysics. The temperature rises by about 38 K after two hours of work. According to the formula, the frequency change in the cavity is about 2 MHz, which will cause jitter in matching the electric field with the electronic pulse and affect the compression effect.

The theoretical operating frequency of our designed cavity is 2.970 GHz, and there may be deviations in the machining and manufacturing process to cause changes in the frequency of the cavity. Since the effect of temperature on frequency is relatively significant and a linear process, we can change the temperature of the water cooling to create a constant water field to change the temperature of the cavity and obtain the ideal operating frequency.

Specifically, O.J. Luiten’s research verified the consistency of SUPERFISH software simulation results with experimental results. We use SUPERFISH software and multiphysics coupling simulation to perform cavity design calculations with the same parameters, and the results are highly consistent.

O.J. Luiten’s research group used SUPERFISH two-dimensional software to simulate and verify similar research work. The comparison between the experimental and simulation results is shown in Figure 5.4 of Reference [32]. The results show that the cavity designed using SUPERFISH software is consistent with the experiment.

We use multiphysics 3D simulation software for electromagnetic–heat coupling analysis and SUPERFISH software to verify the designed resonant cavity. The results shown in Figure 4b and Figure 5b are obtained by comparing the key parameters. When r = 0, the trend in Ez/Ezmax is the same, demonstrating the feasibility and accuracy of designing RF cavities for UED using our simulation method.

To date, there has been no research on electromagnetic–thermal coupling in the simulation of RF compression cavities in UED; our work fills this gap. We also designed a unique cooling structure to ensure the cavity’s temperature stability and provide design ideas for other researchers.

## 5. Conclusions

RF cavities are used in ultrafast electron diffraction systems to improve the temporal resolution. From particle simulation software GPT simulation of the compression effect during the electron leap, we determine that the cavity electric field strength needs 6.5 MV/m. The repetition frequency of our fs laser of 110 MHz determines that the cavity’s resonant frequency should be 2970 MHz. Based on these two indicators, we design the RF compression cavity. In addition, we also use COMSOL Multiphysics to simulate the heat generated by the RF cavity during its operation and its influence on the cavity frequency. Furthermore, we design a water-cooling system to control the cavity’s temperature by changing the water bath’s temperature, which will fine-tune the frequency of the RF cavity and finally achieve the operating frequency. We apply CST, COMSOL, and SUPERFISH, respectively, to perform the simulation calculations of the cavity. To ensure the consistency of the calculation results when performing the calculations, the dimensions of the RF cavity, the boundary conditions, and the parameters of the operating modes (TM_010_) are entirely consistent. This work provides theoretical and experimental support for how to improve temporal resolution. For UED, optimization to achieve single-shot imaging provides a feasible solution.

## Figures and Tables

**Figure 1 sensors-23-07455-f001:**
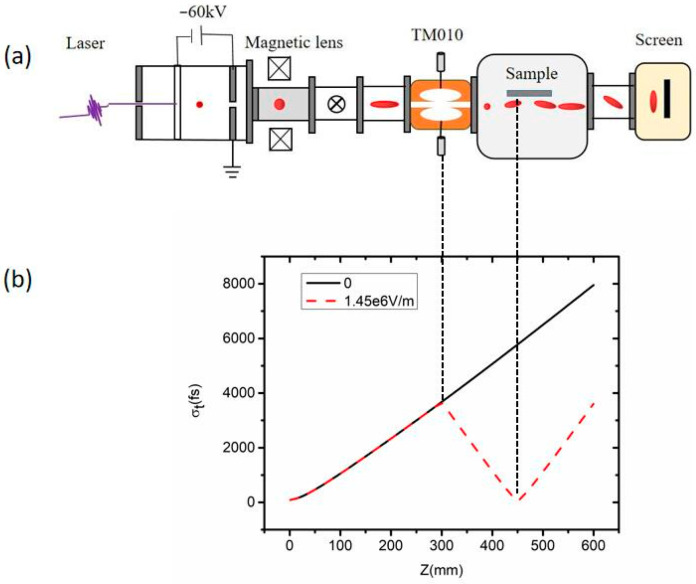
DC ultrafast electronic RF compression system schematic diagram. (**a**) The overall schematic diagram of the electron gun. (**b**) Simulation analysis of longitudinal pulse width variation in the electron beam in the propagation process.

**Figure 2 sensors-23-07455-f002:**
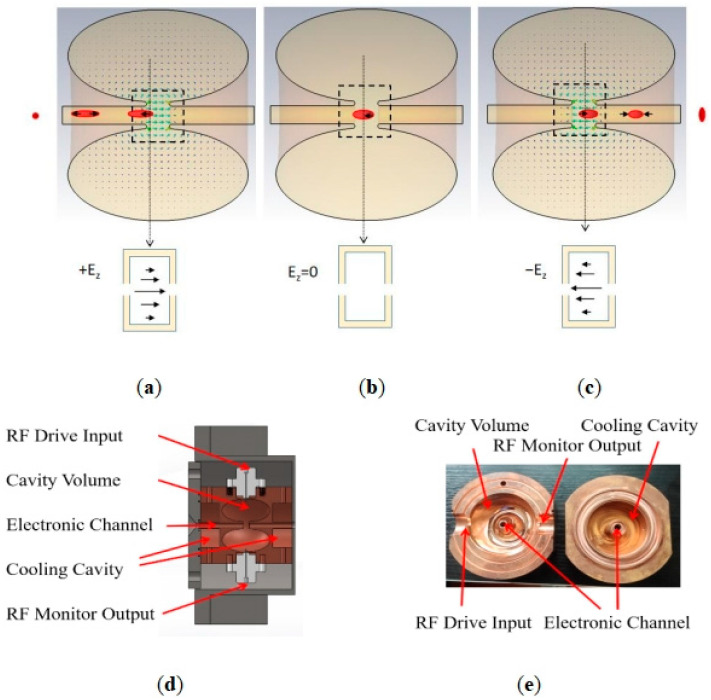
RF cavity and its working principle diagram. (**a**–**c**) correspond to the three states of operation of the RF cavity. (**d**) shows SOLIDWORKS’s 3D design schematic of the RF compression cavity. (**e**) shows the physical diagram of the RF compression cavity.

**Figure 3 sensors-23-07455-f003:**
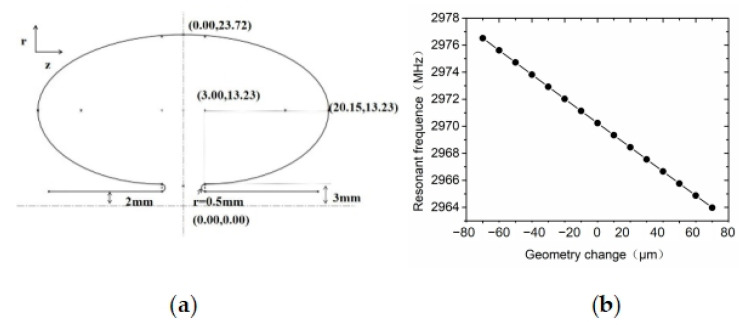
Optimized design of RF compression cavity dimensions. (**a**) shows the cavity dimensions. (**b**) shows the relationship between the resonant frequencies by changing the cavity dimensions.

**Figure 4 sensors-23-07455-f004:**
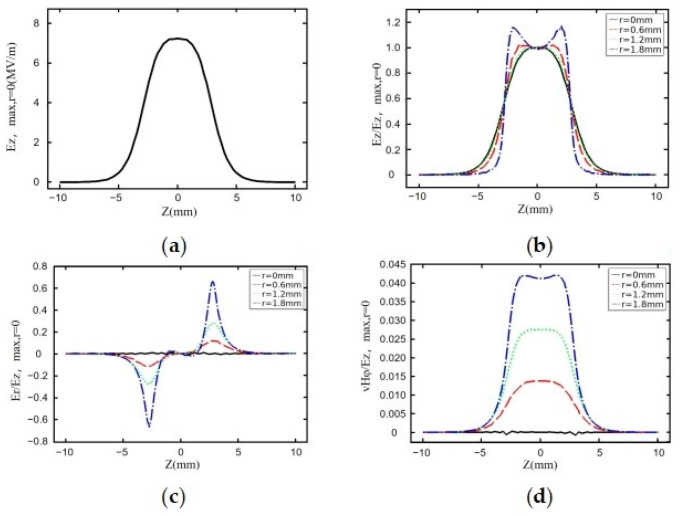
Electromagnetic field distribution in the RF cavity with COMSOL Multiphysics. (**a**) shows the evolution of the electric field. (**b**) shows the ratio of Ez, the electric field in the z-direction at different r positions to Ezmax. (**c**) shows the variation trend of Er, the electric field component in the r-direction. (**d**) shows the variation in the magnetic field force perpendicular to the direction of electron propagation in the working area.

**Figure 5 sensors-23-07455-f005:**
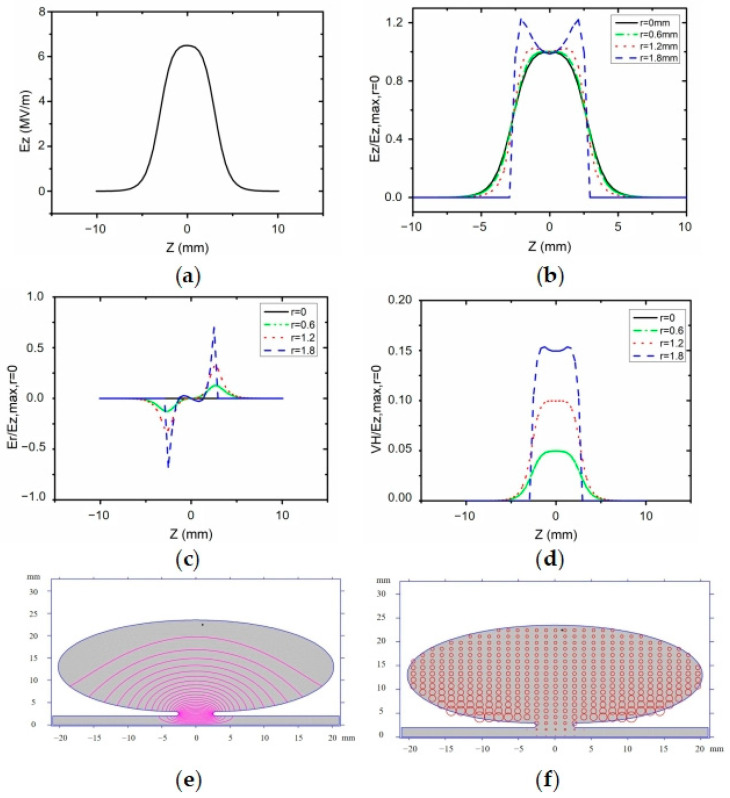
Electromagnetic field distribution in the RF cavity with POISSON SUPERFISH. (**a**) shows the evolution of the electric field. (**b**) shows the ratio of Ez, the electric field in the z-direction at different r positions to Ezmax. (**c**) shows the variation trend of Er, the electric field component in the r-direction. (**d**) shows the variation in the magnetic field force perpendicular to the direction of electron propagation in the working area. (**e**) shows the electric field strength distribution in the RF cavity. (**f**) shows the magnetic field strength distribution in the RF cavity.

**Figure 6 sensors-23-07455-f006:**
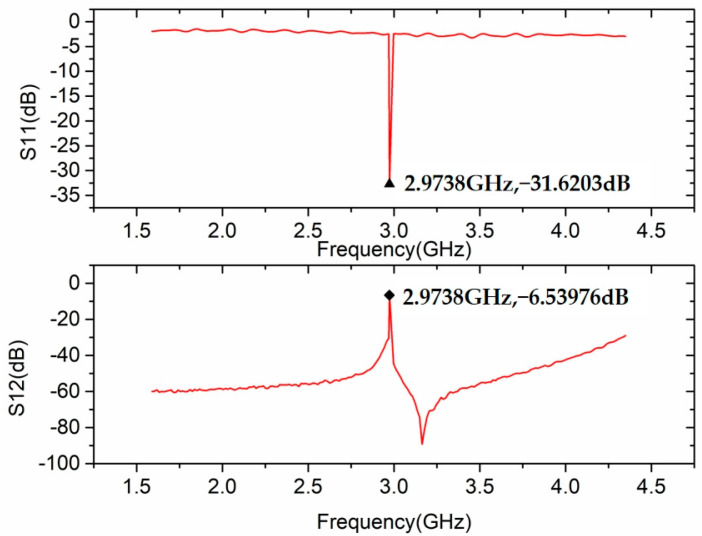
The resonant frequency test of RF compression cavity.

**Figure 7 sensors-23-07455-f007:**
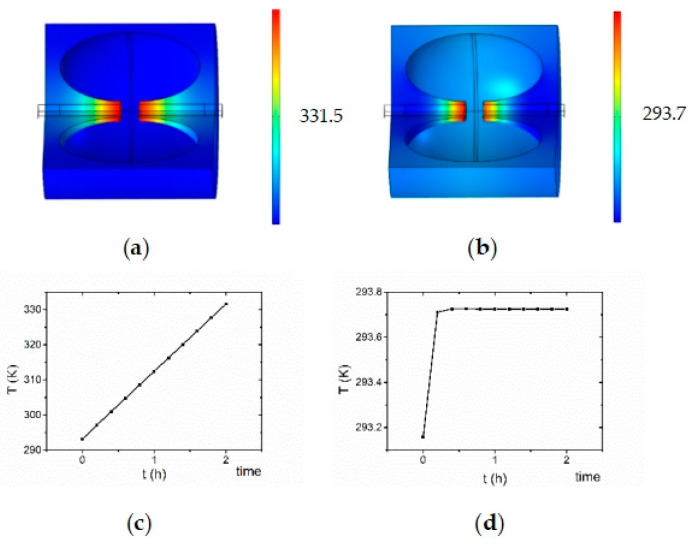
Water-cooling system and the effect of water-cooling system on the operating temperature of RF compression cavity. (**a**,**b**) show three-dimensional simulations of temperature variations. (**c**,**d**) show the changes in the RF compression cavity’s working temperature before and after adding the water-cooling system.

**Table 1 sensors-23-07455-t001:** Comparison of COMSOL Multiphysics and POSSION SUPERFISH results.

	COMSOL Multiphysics	Possion SUPERFISH
Ez, max, r = 0	7.30 MV/m	7.20 MV/m
Ez, max, r = 1.8	8.76 MV/m	8.60 MV/m
Ez, max, r = 1.8	5.00 MV/m	4.90 MV/m
vHφ, max, r = 1.8	0.29 A/m	0.28 A/m
Frequency	2.97035 GHz	2.97065 GHz

## Data Availability

Data availability statement: All relevant data are within the paper. No new data were created or analyzed in this study. Data sharing is not applicable to this article.

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
