# Peer review of "Electromagnetic-Thermal Coupling Study for RF Compression Cavity Applied to Ultrafast Electron Diffraction"

_sensors, 2023, doi:10.3390/s23177455_

Round 1

Reviewer 1 Report

Ultrafast electron diffraction (UED) has high temporal and spatial resolution characteristics, a powerful tool for studying transient structural changes and ultrafast structural dynamics. However, due to the Coulomb repulsion inside an electron pulse, the pulse duration increases significantly, affecting temporal resolution improvement. For this challenge, the authors design a resonant cavity to provide a transiently varying electric field to compress the electron beam and conduct an exploratory analysis of electromagnetic-thermal coupling.

The authors study the electromagnetic-thermal coupling of RF resonators by multiphysics simulation methods and design a low-power resonant cavity with a specific resonant frequency, which provides a feasible way to precisely suppress the broadening of ultrashort electron pulses to improve the temporal resolution. Meanwhile, a water-cooling system is designed to reduce heat accumulation during the RF cavity works to ensure temperature stability.

But there is still a need for modifies in the manuscript, such as

(1)Figure 3(a) is not clear, and the font size is on the small side;

(2) Table 1 does not conform to scientific paper specifications;

(3)More references may need to be cited.

In summary, the manuscript utilizes the electromagnetic-thermal coupling simulation method to carry out innovative research on ultrafast electronic pulse compression cavities, which provides a valuable reference for researchers in the field of UED.

Moderate editing of English language required

Author Response

Thanks for your comments and support on our manuscript “Electromagnetic-Thermal Coupling Study for RF Compression Cavity Applied in Ultrafast Electron Diffraction” (ID: sensors-2519539). We have studied the comments carefully and have made corrections which we hope meet with approval. Revised portions are marked by the “Track Changes” function in MS Word. And we have responded to all the reviewer’s comments.

Editor’s/Reviewers’ comments:

Reviewer #1:

1) Figure 3(a) is not clear, and the font size is on the small side;

Thank you for your suggestions. We have made changes in the latest manuscript submission:

Response 1): Redraw Figure 3(a) and enlarge the font.

Page 8, Line 274-275.

Figure 3 Optimised design of RF compression cavity dimensions

2) Table 1 does not conform to scientific paper specifications;

Response 2): Table 1 has been readjusted as follows to the requirements of the scientific paper.

Page 11, Line374-389.

Table 1 Comparison of COMSOL Multiphysics and POSSION SUPERFISH results

COMSOL Multiphysics

POSSION SUPERFISH

Ez, max, r = 0

7.30 MV/m

7.20 MV/m

Ez, max, r = 1.8

8.76 MV/m

8.60 MV/m

Ez, max, r = 1.8

5.00 MV/m

4.90 MV/m

vHφ, max, r = 1.8

0.29 A/m

0.28 A/m

Frequency

2.97035 GHz

2.97065 GHz

3) More references may need to be cited.

Response 3): For the suggestion of insufficient references you mentioned, we have added references about UED, such as references [1-3] [5-6] [8-10] [12].

Page 1-2, Line 41-51.

After decades of research and development, ultrafast electron diffraction (UED) has provided us with the technical tools to study ultrafast phenomena[1-3]. Compared to conventional electron diffraction tools, UED incorporates temporal resolution, which combines the ultra-high spatial resolution of electron diffraction with the temporal requirements for studying ultrafast dynamics. The application of UED has contributed to many discoveries of ultrafast phenomena in physics[4-6], chemistry[7-10], and biology[11]. It has become a powerful tool for revealing the structural dynamics of molecules and materials at the atomic level.

Page 2, Line 57-59.

For example, some research groups have used compact electron cavities[12-18]to reduce the size of the electron cavity as much as possible, thus reducing the transmission time.

Page 2, Line 80-82.

Based on Siwick’s theory, applying an RF compression cavity to compress the longitudinal pulse width of ultrafast electrons is studied[12].

Reviewer 2 Report

I cannot recommend the publication of your manuscript in its current form. My decision is based on the following key points:

Lack of clarity regarding the nature of the work: In the abstract, it is essential to clearly state whether the study is purely based on simulation or if it includes experimental work. If your work is solely a simulation study, it is important not to mislead the reviewers by implying the presence of experimental data.

Insufficient experimental validation: If your work claims to be an experimental study, it is expected to provide substantial experimental data to support your conclusions. While simulations can be useful for theoretical exploration, they should not replace the need for experimental verification.

Inadequate comparison with existing literature: Your manuscript fails to adequately discuss and compare your results with the existing literature in the field. It is crucial to demonstrate how your work contributes to the current state of knowledge and how it surpasses or complements the existing research.

English expression can meet the basic requirements.

Author Response

Response to Reviewer 2

Thanks for your comments and support on our manuscript “Electromagnetic-Thermal Coupling Study for RF Compression Cavity Applied in Ultrafast Electron Diffraction” (ID: sensors-2519539). We have studied the comments carefully and have made corrections which we hope meet with approval. Revised portions are marked by the “Track Changes” function in MS Word. And we have responded to all the reviewer’s comments.

Editor’s/Reviewers’ comments:

Reviewer #2:

1) Lack of clarity regarding the nature of the work: In the abstract, it is essential to clearly state whether the study is purely based on simulation or if it includes experimental work. If your work is solely a simulation study, it is important not to mislead the reviewers by implying the presence of experimental data.

Response 1): Thank you for your valuable suggestions. Your guidance has been very helpful in illustrating our research work more accurately! First of all, we apologize for not accurately stating the research methodology. We have meticulously revised the abstract and emphasized the finite element(FE) method.

Page 1, Line 16-36.

Abstract

Ultrafast electron diffraction (UED) is a powerful tool for observing the evolution of transient structures at the atomic level. However, temporal resolution is a huge challenge for UEDs, mainly depending on the pulse duration. Unfortunately, the Coulomb force between electrons causes the pulse duration to increase continually when propagating, reducing the temporal resolution. We theoretically design a radio frequency (RF) compression cavity using the finite element method of electromagnetic-thermal coupling to overcome the limitation and obtain a high brightness, short pulse duration, and stable electron beam. In addition, the cavity’s size parameters are optimized, and a water cooling system is designed to ensure stable operation. We adopt the electromagnetic-thermal coupling method for the first time to study the RF cavity applied to UED innovatively. The results show that the RF cavity operates in TM010 mode with a resonant frequency of 2970MHz and generates a resonant electric field. This mode of operation generates an electric field that varies periodically and transiently, compressing the electronic pulse duration. The electromagnetic-thermal coupling method proposed in this study effectively improves the temporal resolution of UED.

To your suggestion 1), we have revised the manuscript’s text, too.

Page 7, Line 256-258.

Therefore, we design the cavity for the first time using COMSOL Multiphysics to obtain the resonant frequency, electric field intensity, and quality factor to meet the design requirements.

Page 7, Line 266-269.

Using COMSOL Multiphysics, we can simulate the resonant frequency, electric field strength, quality factor, and the RF compression cavity’s electric, magnetic, and thermal.

Page 14, Line 488-490.

In addition, we also use COMSOL Multiphysics to simulate the heat generated by the RF cavity during its operation and its influence on the cavity frequency.

2) Insufficient experimental validation: If your work claims to be an experimental study, it is expected to provide substantial experimental data to support your conclusions. While simulations can be useful for theoretical exploration, they should not replace the need for experimental verification.

Response 2): Thanks for your suggestions! The experiments in this study are quite challenging and costly, and sufficient theoretical simulation in the field of UED is a prerequisite for experimental verification. We publish the simulation results first so that the readers can make suggestions, which is conducive to improving the efficiency and level of experimental research and reducing the cost of trial and error.

Based on the simulation, our experiment has completed the cavity’s processing and manufacturing and the resonant frequency test; the test results are shown in the figure below. The experimental resonant frequency is 2.9738GHz, consistent with the simulation with a 0.0038GHz difference. In the future, we will focus on the test of compressed pulse duration, and we need to build a photoelectric synchronous control system and an intact UED system. After our system is built, the entire test results will be published.

Partial experimental results

3) Inadequate comparison with existing literature: Your manuscript fails to adequately discuss and compare your results with the existing literature in the field. It is crucial to demonstrate how your work contributes to the current state of knowledge and how it surpasses or complements the existing research.

Response 3): For your suggestion 3) that the insufficient comparison with the existing literature mentioned is mainly because different UED systems require different resonant frequencies and electric field strengths to compress the pulse duration.

Based on your guidance, we present the accuracy of the conclusions from the following aspects. Specifically, O.J. Luiten’s research verified the consistency of SUPERFISH software simulation results with experimental results. We use SUPERFISH software and multiphysics coupling simulation to perform cavity design calculations with the same parameters, and the results are highly consistent.

O.J. Luiten’s research group used SUPERFISH two-dimensional software to simulate and verify similar research work. The comparison between the experimental and simulation results is shown in Figure 5.4 of Reference [32]. The results show that the cavity designed using SUPERFISH software is consistent with the experiment.

We use multiphysics 3D simulation software for electromagnetic-heat coupling analysis and superfish software to verify the designed resonant cavity. The results shown in Figure 4(b) and Figure 5(b) are obtained by comparing the key parameters. When r=0, the trend of Ez/Ezmax is the same, demonstrating the feasibility and accuracy of designing RF cavities for UED by our simulation method.

See Page 13-14, Line 457-479 in the text.

Figure 5.4: On-axis electric field profile of the compression cavity as calculated with superfish (solid line) and as measured with the perturbation method (dots). The measurement error is very small and can be neglected. Therefore it is not shown. At |z| & 5mm the resolution of the network analyzer is reached, i.e., the frequency shift can not be resolved and a digitalization error becomes apparent[32].

Figure 4(b) shows the ratio of Ez, the electric field in the Z direction at different r positions to Ezmax, and the maximum value of Ez at r=0 with COMSOL Multiphysics.

See Page 9, Line 307-309 in the text.

Figure 5(b). Electromagnetic field distribution in the RF cavity calculated by POSSION SUPERFISH.

See Page 10, Line 346-348 in the text.

There is no research on electromagnetic-thermal coupling in the simulation of RF compression cavities in UED, and our work fills this gap. Meanwhile, we design a unique cooling structure to ensure the cavity’s temperature stability and provide design ideas for other researchers.

See Page 14, Line 475-479 in the text.

Reviewer 3 Report

The Coulomb interactions between electrons cause the electron beam to diverge as it propagates. In this article, a radio-frequency compression cavity is implemented to solve this problem. In addition, the cavity is optimized by a few other ways, such as laser cooling. 

The challenge is the divergence of the electron beam. Some work has been done by other groups to face this challenge. RF has been used in compressing electrons (Appl. Phys. Lett. 101, 081901 (2012) ) and microwave cavity has been used as cited. However, the particular frequency hasn’t been studied experimentally, so the topic is original.

Experimentally testing the RF effects on confining the electron beam with a particular frequency adds new knowledge to the subject area compared to other published material.

Regarding the methodology, further stabilization of the cavity using electronic will be helpful.

The references are cited properly. More references may need to be cited.

Figure 2(e) is not symmetrical. Please explain. It seems that there is a hole on top side of the left part of the same figure. What is the purpose of that hole?

This article is very well written. The only comment that I have is that this article used quite a few softwares. Many of them have hidden parameters. I encourage the authors to develop simple programs to better understand the physics. 

Author Response

Response to Reviewer 3

Thanks for your comments and support on our manuscript “Electromagnetic-Thermal Coupling Study for RF Compression Cavity Applied in Ultrafast Electron Diffraction” (ID: sensors-2519539). We have studied the comments carefully and have made corrections which we hope meet with approval. Revised portions are marked by the “Track Changes” function in MS Word. And we have responded to all the reviewer’s comments.

Editor’s/Reviewers’ comments:

Reviewer #3:

1) The references are cited properly. More references may need to be cited.

Thank you for your suggestions. We have made changes in the latest manuscript submission:

Response 1): For the suggestion of insufficient references you mentioned, we have added references about UED, such as references [1-3] [5-6] [8-10] [12].

Page 1-2, Line 41-51.

After decades of research and development, ultrafast electron diffraction (UED) has provided us with the technical tools to study ultrafast phenomena[1-3]. Compared to conventional electron diffraction tools, UED incorporates temporal resolution, which combines the ultra-high spatial resolution of electron diffraction with the temporal requirements for studying ultrafast dynamics. The application of UED has contributed to many discoveries of ultrafast phenomena in physics[4-6], chemistry[7-10], and biology[11]. It has become a powerful tool for revealing the structural dynamics of molecules and materials at the atomic level.

Page 2, Line 57-59.

For example, some research groups have used compact electron cavities[12-18]to reduce the size of the electron cavity as much as possible, thus reducing the transmission time.

Page 2, Line 80-82.

Based on Siwick’s theory, applying an RF compression cavity to compress the longitudinal pulse width of ultrafast electrons is studied[12].

2) Figure 2(e) is not symmetrical. Please explain. It seems that there is a hole on top side of the left part of the same figure. What is the purpose of that hole?

Response 2): This hole is used for positioning and alignment during welding, but the position has no special requirements and therefore is not symmetrical.

Page 6, Line 211-215.

Figure 2(e) shows the physical diagram of the RF compression cavity, and the asymmetric holes in the Figure are used for positioning and alignment during welding, but there are no specific requirements for their location.

3) This article is very well written. The only comment that I have is that this article used quite a few softwares. Many of them have hidden parameters. I encourage the authors to develop simple programs to better understand the physics.

Response 3): In this study, COMSOL multiphysics simulation is mainly applied for coupling analysis of electromagnetic-heat analysis, SUPERFISH is used to verify the accuracy of simulation results, and CST software is used to illustrate the operating mode of the RF cavity better.

Thank you for your suggestions. In the revised manuscript, we used CST, COMSOL, and SUPERFISH to simulate the cavity. To ensure the consistency of results, the size, boundary conditions, and operating mode (TM010) parameters of the RF cavity are entirely consistent.

See Page 14, Line 496-499 in the text.

In the follow-up research, we will revise the existing theoretical models and summarize the empirical formulas to show the physical process better.

Round 2

Reviewer 2 Report

The author has made appropriate revisions as requested, and the current version can be accepted.

Author Response

Thank you for your valuable suggestions. Your guidance has been very helpful in illustrating our research work more accurately!